# Automating Bayesian optimization with Bayesian optimization

**Gustavo Malkomes, Roman Garnett**
Department of Computer Science and Engineering
Washington University in St. Louis
St. Louis, MO 63130
{luizgustavo, garnett}@wustl.edu

## Abstract

Bayesian optimization is a powerful tool for global optimization of expensive functions. One of its key components is the underlying probabilistic model used for the objective function $f$. In practice, however, it is often unclear how one should appropriately choose a model, especially when gathering data is expensive. We introduce a novel *automated Bayesian optimization* approach that dynamically selects promising models for explaining the observed data using *Bayesian optimization in model space*. Crucially, we account for the uncertainty in the choice of model; our method is capable of using multiple models to represent its current belief about $f$ and subsequently using this information for decision making. We argue, and demonstrate empirically, that our approach automatically finds suitable models for the objective function, which ultimately results in more-efficient optimization.

## 1 Introduction

Global optimization of expensive, potentially gradient-free functions has long been a critical component of many complex problems in science and engineering. As an example, imagine that we want to tune the hyperparameters of a deep neural network in a self-driving car. That is, we want to maximize the generalization performance of the machine learning algorithm, but the functional form of the objective function $f$ is *unknown* and even a single function evaluation is *costly* — it might take hours (or even days!) to train the network. These features render the optimization particularly difficult.

Bayesian optimization has nonetheless shown remarkable success on optimizing expensive gradient-free functions [8, 1, 18]. Bayesian optimization works by maintaining a probabilistic belief about the objective function and designing a so-called *acquisition function* that intelligently indicates the most-promising locations to evaluate $f$ next. Although the design of acquisition functions has been the subject of a great deal of research, how to appropriately model $f$ has received comparatively less attention [17], despite being a decisive factor for performance. In fact, this was considered *the most important problem* in Bayesian optimization by Močkus [12], in a seminal work in the field:

> "The development of some system of *a priori* distributions suitable for different classes of the function $f$ is probably the most important problem in the application of [the] Bayesian approach to ... global optimization" (Močkus 1974, p. 404).

In this work, we develop a search mechanism for appropriate surrogate models (prior distributions) to the objective function $f$. Inspired by Malkomes et al. [11], our model-search procedure operates via Bayesian optimization in model space. Our method does not prematurely commit to a single model; instead, it uses several models to form a belief about the objective function and plan where the next evaluation should be. Our adaptive model averaging approach accounts for model uncertainty, which more realistically copes with the limited information available in practical Bayesian optimization

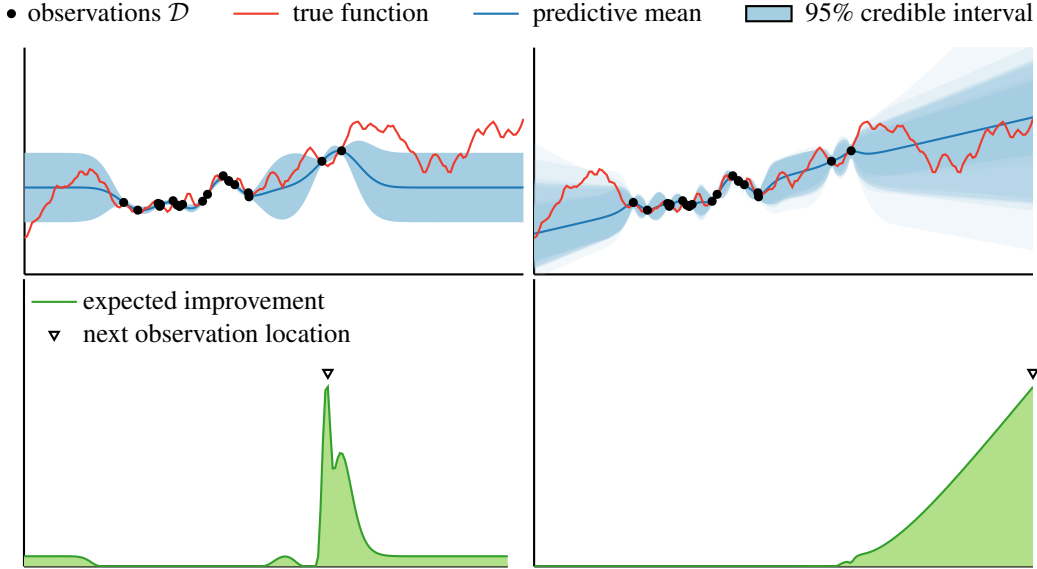

Figure 1: Importance of model selection in Bayesian optimization. Top left: one model represents the belief about the objective. Top right: a mixture of models selected by our approach represents the belief about $f$. Bottom: the acquisition function value (expected improvement) computed using the respective beliefs about the objective. ABO places the next observation at the optimum.

applications. In Figure 1, we show two instances of Bayesian optimization where our goal is to maximize the red objective function $f$. Both instances use expected improvement as acquisition function. The difference between is the belief about $f$: using a single model (left) or combining several models using our automated Bayesian optimization (ABO) approach (right). A single model does not capture the nuances of the true function. In contrast, ABO captures the linear increasing trend of the true function and produces a credible interval which successfully captures the function's behavior. Consequently, ABO finds the optimum in the next iteration.

Finally, we demonstrate empirically that our approach is consistently competitive with or outperforms other strong baselines across several domains: benchmark functions for global optimization functions, hyperparameter tuning of machine learning algorithms, reinforcement learning for robotics, and determining cosmological parameters of a physical model of the Universe.

## 2 Bayesian optimization with multiple models

Suppose we want to optimize an expensive, perhaps black-box function $f\colon \mathcal{X} \to \mathbb{R}$ on some compact set $X \subseteq \mathcal{X}$. We may query $f$ at any point $\mathbf{x}$ and observe a possibly noisy value $y = f(\mathbf{x}) + \varepsilon$. Our ultimate goal is to find the global optimum:

$$\mathbf{x}_{\text{OPT}} = \arg\min_{\mathbf{x} \in X} f(\mathbf{x}) \tag{1}$$

through a sequence of evaluations of the objective function $f$. This problem becomes particularly challenging when we may only make a limited number of function evaluations, representing a real-world *budget B* limiting the total cost of evaluating $f$. Throughout this text, we denote by $\mathcal{D}$ a set of gathered observations $\mathcal{D} = (\mathbf{X}, \mathbf{y})$, where $\mathbf{X}$ is a matrix aggregating the input variables $\mathbf{x}_i \in X$, and $\mathbf{y}$ is the respective vector of observed values $y_i = f(\mathbf{x}_i) + \varepsilon$.

**Modeling the objective function**. Assume we are given a prior distribution over the objective function $p(f)$ and, after observing new information, we have means of updating our belief about $f$ using Bayes' rule:

$$p(f \mid \mathcal{D}) = \frac{p(\mathcal{D} \mid f)p(f)}{p(\mathcal{D})}. \tag{2}$$

The posterior distribution above is then used for decision making, i.e., selecting the $\mathbf{x}$ we should query next. When dealing with a single model, the posterior distribution (2) suffices. Here, however,

we want to make our model of $f$ more flexible, accounting for potential misspecification. Suppose we are given a collection of probabilistic models $\{\mathcal{M}_i\}$ that offer plausible explanations for the data. Each model $\mathcal{M}$ is a set of probability distributions indexed by a parameter $\theta$ from the corresponding model's parameter space $\Theta_{\mathcal{M}}$. With multiple models, we need a means of aggregating their beliefs. We take a fully Bayesian approach and we use the *model evidence* (or *marginal likelihood*), the probability of generating the observed data given a model $\mathcal{M}$,

$$p(\mathbf{y} \mid \mathbf{X}, \mathcal{M}) = \int p(\mathbf{y} \mid \mathbf{X}, \theta, \mathcal{M})\, p(\theta \mid \mathcal{M})\, \mathrm{d}\theta, \tag{3}$$

as the key quantity for measuring the fit of each model to the data. The evidence integrates over the parameters $\theta$ to compute the probability of the model generating the observed data under a hyperprior distribution $p(\theta \mid \mathcal{M})$. Given (3), one can easily compute the *model posterior*,

$$p(\mathcal{M} \mid \mathcal{D}) = \frac{p(\mathbf{y} \mid \mathbf{X}, \mathcal{M})p(\mathcal{M})}{p(\mathbf{y} \mid \mathbf{X})} = \frac{p(\mathbf{y} \mid \mathbf{X}, \mathcal{M})p(\mathcal{M})}{\sum_i p(\mathbf{y} \mid \mathbf{X}, \mathcal{M}_i)p(\mathcal{M}_i)}, \tag{4}$$

where $p(\mathcal{M})$ represents a prior probability distribution over the models. The model posterior gives us a principled way of combining the beliefs of all models. Our model of $f$ can now be summarized with the following model-marginalized posterior distribution:

$$p(f \mid \mathcal{D}) = \sum_i p(\mathcal{M}_i \mid \mathcal{D}) \underbrace{\int p(f \mid \mathcal{D}, \theta, \mathcal{M}_i)p(\theta \mid \mathcal{D}, \mathcal{M}_i)\, \mathrm{d}\theta}_{p(f \mid \mathcal{D}, \mathcal{M}_i)}. \tag{5}$$

Note that (5) takes into consideration all plausible models $\{\mathcal{M}_i\}$ and the integral $p(f \mid \mathcal{D}, \mathcal{M}_i)$ accounts for the uncertainty in each model's hyperparameters $\theta \in \Theta_{\mathcal{M}_i}$. Unfortunately, the latter is often intractable and we will discuss means of approximating it in Section 4.2. Next, we describe how to use the model-marginalized posterior to intelligently optimize the objective function.

**Selecting where to evaluate next**. Given our belief about $f$, we want to use this information to select which point $\mathbf{x}$ we want to evaluate next. This is typically done by maximizing an *acquisition function* $\alpha\colon \mathcal{X} \to \mathbb{R}$. Instead of solving (1) directly, we optimize the proxy (and simpler) problem

$$\mathbf{x}^* = \underset{\mathbf{x} \in X}{\arg\max}\, \alpha(\mathbf{x}; \mathcal{D}). \tag{6}$$

We use *expected improvement* (EI) [12] as our acquisition function. Suppose that $f'$ is the minimal value observed so far.[1] EI selects the point $\mathbf{x}$ that, in expectation, improves upon $f'$ the most:

$$\alpha_{\mathrm{EI}}(\mathbf{x}; \mathcal{D}, \mathcal{M}) = \mathbb{E}_y\big[\max(f' - y, 0) \mid \mathbf{x}, \mathcal{D}, \mathcal{M}\big]. \tag{7}$$

Note that if $p(y \mid \mathbf{x}, \mathcal{D}, \mathcal{M})$ is a Gaussian distribution (or can be approximated as one), the expected improvement can be computed in closed form. Usually, acquisition functions are evaluated for a given model choice $\mathcal{M}$. As before, we want to incorporate multiple models in this framework. For EI, we can easily take into account all models as follows:

$$\alpha_{\mathrm{EI}}(\mathbf{x}; \mathcal{D}) = \mathbb{E}_{y, \mathcal{M}}\big[\max(f' - y, 0) \mid \mathbf{x}, \mathcal{D}\big] = \mathbb{E}_{\mathcal{M}}\big[\alpha_{\mathrm{EI}}(\mathbf{x}; \mathcal{D}, \mathcal{M})\big]. \tag{8}$$

We could also derive similar results for other acquisition functions such as probability of improvement [9] and GP upper confidence bound (GP-UCB) [19].

## 3 Automated model selection for fixed-size datasets

Before introducing our automated method for Bayesian optimization, we need to review a previously proposed method for automated model selection of fixed-size datasets. We begin with a brief introduction to Gaussian processes and a description of the model space we adopted in this paper.

**Gaussian processes models**. We take a standard nonparametric approach and place a Gaussian process (GP) prior distribution on $f$, $p(f) = \mathcal{GP}(f; \mu, K)$, where $\mu\colon \mathcal{X} \to \mathbb{R}$ is a mean function and $K\colon \mathcal{X} \times \mathcal{X} \to \mathbb{R}$ is a positive-semidefinite covariance function or kernel. Both $\mu$ and $K$ may have hyperparameters, which we conveniently concatenate into a single vector $\theta$. To connect to our

framework, a GP model $\mathcal{M}$ comprises $\mu$, $K$, and a prior over its associated hyperparameters $p(\theta)$. Thanks to the elegant marginalization properties of the Gaussian distribution, computing the posterior distribution $p(f \mid \theta, \mathcal{D})$ can be done in closed form, if we assume a standard Gaussian likelihood observation model, $\varepsilon \sim N(0, \sigma^2)$. For a more detailed introduction to GPs, see [16].

Gaussian processes are extremely powerful modeling tools. Their success, however, heavily depends on an appropriate choice of the mean function $\mu$ and covariance function $K$. In some cases, a domain expert might have an informative opinion about which GP model could be more fruitful. Here, however, we want to avoid human intervention and propose an automatic approach.

**Space of models**. First we need a space of GP models that is general enough to explain virtually any dataset. We adopt the generative kernel grammar of [2] due to its ability to create arbitrarily complex models. We start with a set of so-called *base* (one-dimensional) kernels, such as the common squared exponential (SE) and rational quadratic (RQ) kernels. Then, we create new and potentially more complex kernels by summation and multiplication, over individual dimensions, of the base kernels. This let us create kernels over multidimensional inputs. As a result, we have a space of kernels that allows one to search for appropriate structures (different kernel choices) as well as relevant features (subsets of the input). Now, we need an efficient method for searching models from this given space. Fortunately, this was accomplished by the work of [11], which we summarize next.

**Bayesian optimization for model search**. Suppose we are given a space of probabilistic models $\mathbb{M}$ such as the above-cited generative kernel grammar. As mentioned before, the key quantity for model comparison in a Bayesian framework is the model *evidence* (3). Previous work has shown that we can search for promising models $\mathcal{M} \in \mathbb{M}$ by viewing the evidence as a function $g \colon \mathbb{M} \to \mathbb{R}$ to be optimized [11]. Their method consists of a Bayesian optimization approach to model selection (BOMS), in which we try to find the optimal model

$$\mathcal{M}_{\text{OPT}} = \arg\max_{\mathcal{M} \in \mathbb{M}} g(\mathcal{M}; \mathcal{D}), \tag{9}$$

where $g(\mathcal{M}; \mathcal{D})$ is the (log) model evidence: $g(\mathcal{M}; \mathcal{D}) = \log p(\mathbf{y} \mid \mathbf{X}, \mathcal{M})$. Two key aspects of their method deserve special attention: their unusual GP prior, $p(g) = \mathcal{GP}(g; \mu_g, K_g)$, where the mean and covariance functions are appropriately defined over the model space $\mathbb{M}$; and their heuristic for traversing $\mathbb{M}$ by maintaining a set of candidate models $\mathcal{C}$. The precise mechanism for traversing the space of models is not particular relevant for our exposition, but the fact that $\mathcal{C}$ is changing as we search for better models is. Due to limited space, we refer the reader to the original work for more information. Nevertheless, it is important to note that their approach was shown to be more efficient than previous methods.

# 4 Automating Bayesian optimization with Bayesian optimization

Here, we present our automated Bayesian optimization (ABO) algorithm. ABO is a two-level Bayesian optimization procedure. The "outer level" solves the standard Bayesian optimization problem, where we want to search for the optimum of the objective function $f$. Inside the Bayesian optimization loop, we use a second "inner" Bayesian optimization, where the goal is to search for appropriate models $\{\mathcal{M}_i\}$ to the objetive function $f$. The inner optimization seeks models maximizing the model evidence as in BOMS (Section 3). The motivation is to refine the set of models $\{\mathcal{M}_i\}$ before choosing where we want to query the (expensive) objective function $f$ next. Given a set of models, we can use the methodology presented in Section 2 to perform Bayesian optimization with multiple models.

In the next subsection, we will describe the inner Bayesian optimization method which we refer to as active BOMS (ABOMS). Before going to the second Bayesian optimization level, we summarize ABO in Algorithm 1. First, we initialize our set of promising models $\{\mathcal{M}_i\}$ with random models chosen from the grammar of kernel, same used in [2]. To select these models, we perform random walks from the the empty kernel and repeatedly apply a random number of grammatical operations. The number of operations is sampled from a geometric distribution with termination probability of $\frac{1}{3}$. Then, at each iteration: we update all models with current data, computing the corresponding model evidence of each model; use ABOMS (the inner model-search optimization) to include more promising candidate models in $\{\mathcal{M}_i\}$; exclude all models that are unlikely to explain the current data, those with $p(\mathcal{M} \mid \mathcal{D}) < 10^{-4}$; sample the function at location $\mathbf{x}^*$ using (8) and all models $\{\mathcal{M}_i\}$; finally, we evaluate $y^* = f(\mathbf{x}^*) + \varepsilon$ and include this new observation in our dataset.

---

**Algorithm 1** Automated Bayesian Optimization

---

   **Input:** function $f$, budget $B$, initial data $\mathcal{D}$
   $\{\mathcal{M}_i\} \leftarrow$ Initial set of promising models
   **repeat**
       $\{\mathcal{M}_i\} \leftarrow$ update models $(\{\mathcal{M}_i\}, \mathcal{D})$
       $\{\mathcal{M}_i\} \leftarrow \textsc{abOMS}(\{\mathcal{M}_i\}, \mathcal{D})$
       $p(\mathcal{M} \mid \mathcal{D}) \leftarrow$ compute model posterior
       discard irrelevant models $p(\mathcal{M}_i \mid \mathcal{D}) < 10^{-4}$
       $\mathbf{x}^* \leftarrow \arg\max_{\mathbf{x} \in X} \alpha_{\text{EI}}(\mathbf{x}\,; \mathcal{D})$.
       $y^* \leftarrow f(\mathbf{x}^*) + \varepsilon$
       $\mathcal{D} \leftarrow \mathcal{D} \cup \{(\mathbf{x}^*, y^*)\}$
   **until** budget $B$ is depleted

---

## 4.1 Active Bayesian optimization for model search

The critical component of ABO is the inner optimization procedure that searches for suitable models to the objective function: the active Bayesian optimization for model search (ABOMS). Notice that the main challenge is that ABOMS is nested in a Bayesian optimization loop, meaning that both *data* and *models* will change as we perform more outer Bayesian optimization iterations.

Suppose we already gathered some observations $\mathcal{D}$ of the objective function $f$. Additionally, we use the previously proposed BOMS (Section 3) as the inner model search procedure. Inside BOMS, we tried different models, gathering observations of the (log) model evidence, $g(\mathcal{M}; \mathcal{D}) = \log p(\mathbf{y} \mid \mathbf{X}, \mathcal{M})$. We denote by $\mathcal{D}_g = \{\mathcal{M}_j, g(\mathcal{M}_j; \mathcal{D})\}$ the observations of the inner Bayesian optimization. After one loop of the outer Bayesian optimization, we obtain new data $\mathcal{D}' = \mathcal{D} \cup \{(\mathbf{x}^*, y^*)\}$. Now, the model evidence of all previously evaluated models $\mathcal{D}_g$ changes since $g(\mathcal{M}_j, \mathcal{D}) \neq g(\mathcal{M}_j, \mathcal{D}')$ for all $j$. As a result, we would have to retrain all models in $\mathcal{D}_g$ to correctly compare them. Recall that there are good models in $\mathcal{D}_g$ for explaining the objective function $f$. These models will be passed to the outer Bayesian optimization, where they will be updated — ultimately, we want to provide outstanding suggestions $\mathbf{x}^*$ for where to query $f$ next, thus they need to be retrained. A *large portion* of the tested models in $\mathcal{D}_g$, however, are not appropriated for modeling $f$; in fact, they can be totally ignored by the outer optimization. Yet these "bad" models can help guide the search toward more-promising regions of model space. How to retain information from previously evaluated models without resorting to exhaustive retraining?

Our answer is to modify BOMS in two ways. First, we place a GP on the *normalized* model evidence, $g(\mathcal{M}; \mathcal{D}) = \log p(\mathbf{y} \mid \mathbf{X}, \mathcal{M}) / |\mathcal{D}|$, which let us compare models across iterations. Second, we assume that each evidence evaluation is *corrupted by noise*, the variance of which depends on the number of data points used to compute it: the more data we use, the more accurate our estimate, and the lower the noise. More specifically, we use the same GP prior of [11], $p(g) = \mathcal{GP}(g; \mu_g, K_g)$, where $\mu_g \colon \mathbb{M} \to \mathbb{R}$ is just a constant mean function and $K_g \colon \mathbb{M}^2 \to \mathbb{R}$ is the "kernel kernel" defined as a squared exponential kernel that uses the (averaged) Hellinger distance between the inputs as oppose to the standard $\ell_2$ norm (see the original paper for more details). Our observation model, however, assumes that the observations of the normalized model evidence are corrupted by heterogenous noise:

$$y_g(\mathcal{M}; \mathcal{D}_n) = \frac{g(\mathcal{M}; \mathcal{D}_n)}{n} + \varepsilon\left(\frac{1}{n}\right). \tag{10}$$

To choose the amount of noise, we observed that, using the chain rule, the marginal likelihood can be written as $\log p(\mathbf{y} \mid \mathbf{X}, \mathcal{M}) = \sum_i \log p\big(y_i \mid \mathbf{x}_i, \{(\mathbf{x}_j, y_j) \mid j < i\}, \mathcal{M}\big)$, which is the sum of the marginal predictive log likelihoods for the points in the $\mathcal{D}$. When we divide $\log p(\mathbf{y} \mid \mathbf{X}, \mathcal{M})$ by $|\mathcal{D}| = n$, we can *interpret* the result as an estimate of the average predictive log marginal likelihood.[2] Therefore, if

$$\log p(\mathbf{y} \mid \mathbf{X}, \mathcal{M})/n \approx \mathbb{E}\big[\log p(y^* \mid \mathbf{x}^*, \mathcal{D}, \mathcal{M}) \mid \mathcal{M}\big],$$

then the variance of this estimate with $n$ measurements is

$$\text{Var}\Big[\log p\big(y_i \mid \mathbf{x}_i, \{(\mathbf{x}_j, y_j) \mid j < i\}, \mathcal{M}\big)\Big]/n.$$

which shrinks like $\sigma_g^2/n$ for a small constant $\sigma_g$ (e.g., 0.5). For large $n$ it goes to $0$. This mechanism gracefully allows us to condition on the history of all previously proposed models during the search. By modeling earlier evidence computations as noisier, we avoid recomputing the model evidence of previous models every round, but we still make the search for good models better informed.

## 4.2 Implementation

In practice, several distributions presented above are often intractable for GPs. Now, we discuss how to efficiently approximate these quantities. First, instead of using just a delta approximation to the hyperparameter posterior $p(\theta \mid \mathcal{D}, \mathcal{M})$, e.g. MLE/MAP, we use a Laplace approximation, i.e., we make a second-order Taylor expansion around its mode: $\hat{\theta} = \arg\max_\theta \log p(\theta \mid \mathcal{D}, \mathcal{M})$. This results in a multivariate Gaussian approximation:

$$p(\theta \mid \mathcal{D}, \mathcal{M}) \approx \mathcal{N}(\theta; \hat{\theta}, \Sigma) \text{ where } \Sigma^{-1} = -\nabla^2 \log p(\theta \mid \mathcal{D}, \mathcal{M})\big|_{\theta=\hat{\theta}}.$$

Conveniently, the Laplace approximation also give us a means of approximating the model evidence:

$$\log p(\mathbf{y} \mid \mathbf{X}, \mathcal{M}) \approx \log p(\mathbf{y} \mid \mathbf{X}, \hat{\theta}, \mathcal{M}) + \log p(\hat{\theta} \mid \mathcal{M}) - \tfrac{1}{2}\log\det\Sigma^{-1} + \tfrac{d}{2}\log 2\pi,$$

where $d$ is the dimension of $\theta$. The above approximation can be interpreted as rewarding explaining the data well while penalizing model complexity [13, 15].

Next consider the posterior distribution $p(f \mid \mathcal{D}, \mathcal{M})$, which is an integral over the model's hyperparameters. This distribution is intractable, even with our Gaussian approximation to the hyperparameter posterior $p(\theta \mid \mathcal{D}, \mathcal{M}) \approx \mathcal{N}(\theta; \hat{\theta}, \Sigma)$. We use a general approximation technique originally proposed by [14] (Section 4) in the context of Bayesian quadrature. This approach assumes that the posterior mean of $p(f^* \mid \mathbf{x}^*, \mathcal{D}, \theta, \mathcal{M})$ is affine in $\theta$ around $\hat{\theta}$ and the GP covariance is constant. Let

$$\mu^*(\theta) = \mathbb{E}[f^* \mid \mathbf{x}^*, \mathcal{D}, \theta, \mathcal{M}] \quad \text{and} \quad \nu^*(\theta) = \text{Var}[f^* \mid \mathbf{x}^*, \mathcal{D}, \theta, \mathcal{M}]$$

be the posterior predictive mean and variance of $f^*$. The result of this approximation is that the posterior distribution of $f^*$ is approximated by

$$p(f^* \mid \mathbf{x}^*, \mathcal{D}, \mathcal{M}) \approx \mathcal{N}\big(f^*; \mu^*(\hat{\theta}), \sigma_{\text{AFFINE}}^2\big), \quad \text{where} \quad \sigma_{\text{AFFINE}}^2 = \big[\nabla\mu^*(\hat{\theta})\big]^\top \Sigma \big[\nabla\mu^*(\hat{\theta})\big].$$

This approach was shown to be a good alternative for propagating the uncertainty in the hyperparameters [14]. Finally, given the Gaussian approximations above (11), we use standard techniques to analytically approximate the predictive distribution:

$$p(y^* \mid \mathbf{x}^*, \mathcal{D}, \mathcal{M}) = \int p(y^* \mid f^*)\, p(f^* \mid \mathbf{x}^*, \mathcal{D}, \mathcal{M})\, \mathrm{d}f^*.$$

Our code and data will be available online: `https://github.com/gustavomalkomes/abo`.

## 5 Related Work

Our approach is inspired by some recent developments in the field of automated model selection [11, 2, 6]. Here, we take these ideas one step further and consider automated model selection when actively acquiring new data.

Gardner et al. [3] also tackled the problem of model selection in an active learning context but with a different goal. Given a *fixed set* of candidate models, the authors proposed a method for gathering data to quickly identify which model best explains the data. Here our ultimate goal is to perform global optimization (1) when we can dynamically change our set of models. In future work, it would be interesting to examine whether it may be possible to combine our ideas with their proposed method to actively learn in model space.

More recently, Gardner et al. [4] developed an automated model search for Bayesian optimization similar to our method. Their approach, however, uses a MCMC strategy for sampling new promising models, whereas we adapt the Bayesian optimization search of proposed by Malkomes et al. [11]. We will discuss further differences between our approach and their MCMC method in the next section.

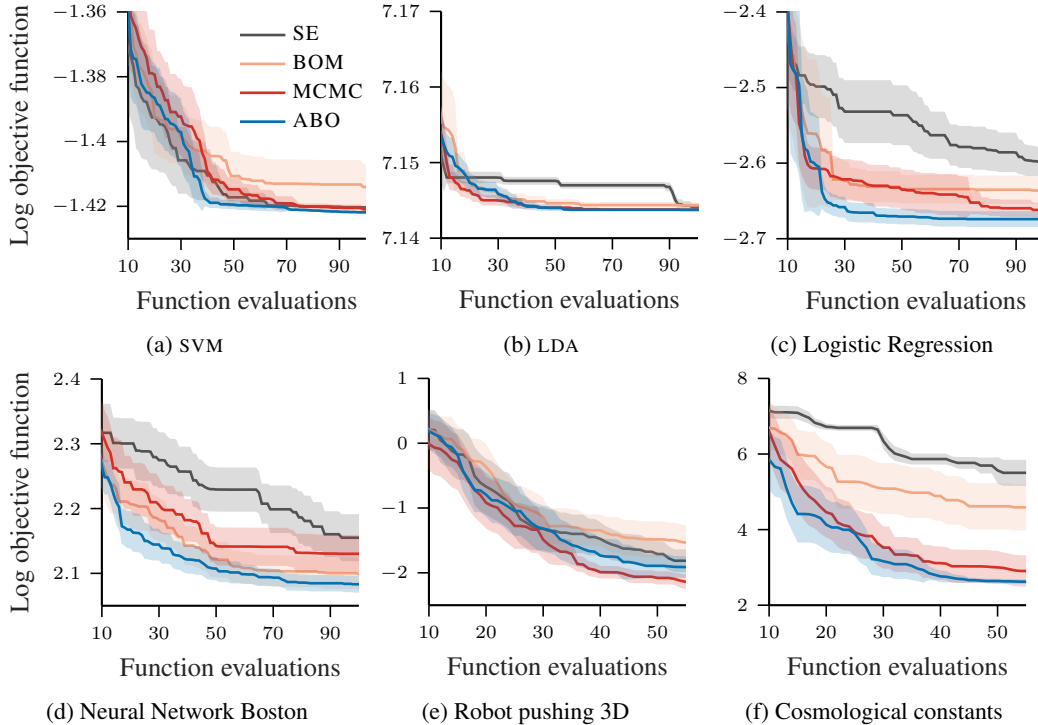

(a) SVM      (b) LDA      (c) Logistic Regression

(d) Neural Network Boston      (e) Robot pushing 3D      (f) Cosmological constants

Figure 2: Averaged minimum observed function value and standard error of all methods for several objective functions. For better visualization, we omit the first 10 function evaluations since they are usually much higher than the final observations.

## 6 Empirical Results

We validate our approach against several optimization alternatives and across several domains. Our first baseline is a random strategy that selects twice as many locations as the other methods. We refer to this strategy as RANDOM $2\times$ [10]. We also consider a competitive Bayesian optimization implementation which uses a single non-isotropic squared exponential kernel (SE), expected improvement as the acquisition function and all the approximations described in Section 4.2. Then, we considered two more baselines that represent the uncertainty about the unknown function through a combination of multiple models. One baseline uses the same collection of predefined models throughout its execution; we refer to this approach as the bag of models (BOM). The other is an adaptation of the method proposed in [4], here referred as MCMC, which, similar to ABO, is allowed to dynamically select more models every iteration. Instead of using the additive class of models proposed in the original work, we adapted their Metropolis–Hastings algorithm to the more-general compositional grammar proposed by Duvenaud et al. [2], which is also used by our method. This choice lets us compare which adaptive strategy performs better in practice. Specifically, given an initial model $\mathcal{M}$, the MCMC proposal distribution randomly selects a neighboring model $\mathcal{M}'$ from the grammar. Then we compute the acceptance probability as in [4].

All multiple models strategies (BOM, MCMC and ABO) start with the same selection of models (See Section 4) and they aim to maximize the model-marginalized expected improvement (8). Both adaptive algorithms (ABO and MCMC) are allowed to perform five model evidence computations before each function evaluation; ABO queries five new models and MCMC performs five new proposals. In our experiments, we limited the number of models to 50, always keeping those with the higher model evidence. Model choice and acquisition functions apart, we kept all configurations the same. All methods used L-BFGS to optimize each model's hyperparameters. To avoid bad local minima, we perform two restarts, each begining from a sample of $p(\theta \mid \mathcal{M})$. All the approximations described in Section 4.2 were also used. We maximized the acquisition functions by densely sampling $1000d^2$ points from a $d$-dimensional low-discrepancy Sobol sequence, and starting MATLAB `fmincon` (a

Table 1: Results for the average gap performance across 20 repetitions for different test functions and methods. RANDOM $2\times$ (R $2\times$) results are averaged across 1000 experiments. Numbers that are *not* significantly different from the highest average gap for each function are bolded (one-sided paired Wilcoxon signed rank test, 5% significance level).

| | function | $d$ | R $2\times$ | SE | BOM | MCMC | ABO |
|---|---|---|---|---|---|---|---|
| synthetic objectives | Ackley $2d$ | 2 | 0.422 | 0.717 | **0.984** | **0.988** | **0.980** |
| | Beale | 2 | 0.725 | 0.541 | **0.644** | **0.596** | **0.688** |
| | Branin | 2 | 0.743 | **1.000** | 0.950 | 0.996 | 0.998 |
| | Eggholder | 2 | 0.461 | **0.516** | **0.529** | **0.546** | **0.579** |
| | Six-Hump Camel | 2 | 0.673 | 0.723 | **0.988** | **0.992** | **0.998** |
| | Drop-Wave | 2 | **0.458** | **0.496** | 0.421 | 0.447 | **0.481** |
| | Griewank $2d$ | 2 | 0.669 | **0.924** | **0.954** | **0.941** | **0.964** |
| | Rastrigin | 2 | 0.538 | 0.410 | **0.832** | **0.827** | **0.850** |
| | Rosenbrock | 2 | 0.787 | **1.000** | 0.999 | 0.993 | 0.999 |
| | Shubert | 2 | 0.337 | **0.384** | **0.374** | **0.332** | **0.481** |
| | Hartmann | 3 | 0.682 | **1.000** | **0.970** | 0.999 | **1.000** |
| | Levy | 3 | 0.669 | 0.774 | **0.913** | 0.942 | **0.971** |
| | Rastrigin $4d$ | 4 | 0.414 | 0.261 | **0.823** | 0.715 | **0.821** |
| | Ackley $5d$ | 5 | 0.299 | 0.736 | 0.409 | **0.886** | 0.809 |
| | Griewank $5d$ | 5 | 0.605 | **0.971** | 0.756 | **0.974** | 0.968 |
| | mean gap | | 0.566 | 0.697 | 0.770 | 0.812 | 0.839 |
| | median gap | | 0.605 | 0.723 | 0.832 | 0.941 | 0.964 |
| real-world objectives | SVM | 3 | 0.903 | 0.912 | 0.840 | 0.938 | **0.956** |
| | LDA | 3 | 0.939 | **0.950** | 0.925 | **0.950** | **0.950** |
| | Logistic regression | 4 | 0.928 | 0.774 | 0.899 | 0.936 | **0.994** |
| | Robot pushing $3d$ | 3 | 0.815 | 0.927 | 0.878 | **0.967** | 0.935 |
| | Robot pushing $4d$ | 4 | **0.824** | **0.748** | 0.619 | **0.668** | **0.715** |
| | Neural network Boston | 4 | 0.491 | 0.594 | 0.703 | 0.640 | **0.757** |
| | Neural network cancer | 4 | **0.845** | **0.645** | **0.682** | **0.773** | **0.749** |
| | Cosmological constants | 9 | 0.739 | 0.848 | 0.859 | 0.984 | **0.999** |
| | mean gap | | 0.810 | 0.800 | 0.801 | 0.857 | 0.882 |
| | median gap | | 0.834 | 0.811 | 0.850 | 0.937 | 0.943 |

local optimizer) from the sampled point with highest value. Each experiment, was repeated 20 times with five random initial examples, which were the same for all Bayesian optimization methods. RANDOM $2\times$ results were averaged across 1000 repetitions.

**Benchmark functions for global optimization.** Our first set of experiments are test functions commonly used as benchmarks for optimization [20]. We adopted a similar setup as previous works [5] but included more test functions. The goal is to find the global minimum of each test function given a limited number of function evaluations. We provide more information about the chosen functions in the supplementary material. The maximum number of function evaluations was limited to 10 times the dimensionality of the function domain being optimized. We report the gap measure [7], defined as $\frac{f(x_{\text{first}}) - f(x_{\text{best}})}{f(x_{\text{first}}) - f(x_{\text{OPT}})}$, where $f(x_{\text{first}})$ is the minimum function value among the first initial random points, $f(x_{\text{best}})$ is the best value found by the method, and $f(x_{\text{OPT}})$ is the optimum.

Table 1 (top) shows the results for different functions and methods. For each test function, we perform a one-sided Wilcoxon signed rank test at the 5% significance level with each method and the one that had the highest average performance. All results that are *not* significantly different than the highest are marked in bold. First, note that RANDOM $2\times$ performs poorly in these synthetic constructed "hard" functions. Then, observe that the overall performance of all multi-model methods is higher than the single GP baseline, with ABO leading these algorithms with respect to the mean and median gap performance over all functions. In fact, ABO's performance is comparable to the best method for 11 out of 15 functions.

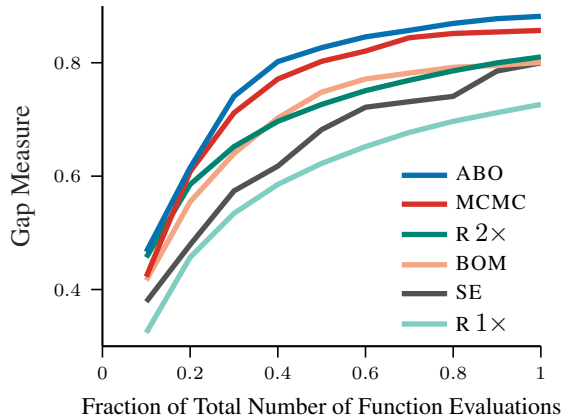

Figure 3: Average gap across the eight real-world objective functions vs. fraction of total number of function evaluations. Here, we display the performance of random search (R 1×) for reference.

**Real-world optimization functions**. To further investigate the importance of model search, we consider a second set of functions used in recent publications. Our goal was to select a diverse and challenging set of functions which might better demonstrate the performance of these algorithms in a real application. More information about these functions is given in the supplementary material.

We show the gap measure for the second set of experiments in Table 1 (bottom) and perform the same statistical test as before. For computing the gap measure in these experiments, when the true global minimum is unknown, we used the minimum observed value across all experiments as a proxy for the optimal value. In Figure 3 we show the average gap measure across all eight test functions as a function of the total number of functions evaluations allowed. In Figure 2, we show the averaged minimum observed function value and standard error of all methods for 6 out of the 8 functions (see supplementary material for the other two functions).

With more practical objective functions, the importance of model search becomes more clear. ABO either outperforms the other methods (4 out of the 8 datasets) or achieves the lowest objective function. Figure 3 also shows that ABO quickly advances on the search for the global minimum — on average, the gap measure is higher than 0.8 after at half of the budget. Interestingly, RANDOM 2× also performs well for 2 out of these 8 datasets, those are the problems in which all methods have a similar performance, suggesting that these functions are easier to optimize than the others.

Naturally, training more models and performing an extra search to dynamically select models require more computation than running a standard single Bayesian optimization. In our implementation, not optimized for speed, the median wall clock across all test functions for *updating* and searching the five new models was 65 and 41 seconds, respectively, for MCMC and ABO. Note that the model update is what dominates this procedure for both methods, with MCMC tending to select more complex models than ABO. In practice, one could perform this step in parallel with the expensive objective function evaluation, requiring no additional overhead besides the cost of optimizing the model-marginal acquisition function, which can also be adjusted by the user.

# 7   Conclusion

We introduced a novel automated Bayesian optimization approach that uses multiple models to represent its belief about an objective function and subsequently decide where to query next. Our method automatically and efficiently searches for better models as more data is gathered. Empirical results show that the proposed algorithm often outperforms the baselines for several different objective functions across multiple applications. We hope that this work can represent a step towards a fully automated system for Bayesian optimization that can be used by a nonexpert on arbitrary objectives.

## Acknowledgments

GM, and RG were supported by the National Science Foundation (NSF) under award number IIA–1355406. GM was also supported by the Brazilian Federal Agency for Support and Evaluation of Graduate Education (CAPES).

## Footnotes

[1]We make a simplifying assumption that the noise level is small, thus $f' \approx \min_i \mu_{f \mid \mathcal{D}}(\mathbf{x}_i)$ and $y(\mathbf{x}) \approx f(\mathbf{x})$.

[2]Note that the training data is not independent since we are choosing the locations $\mathbf{x}$, and we are not assuming that $n \to \infty$

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
