[Supplementary Material]

# Automating Bayesian optimization with Bayesian optimization: supplementary material

**Gustavo Malkomes, Roman Garnett**
Department of Computer Science and Engineering
Washington University in St. Louis
St. Louis, MO 63130
{luizgustavo, garnett}@wustl.edu

## 1 Experiments

In this section, we provide more details about our experiments. First, we give more information about the synthetic test functions that we used in Table 1, then we describe the "real-world" functions.

**Tuning hyperparameters of machine learning algorithms** We consider hyperparameter-tuning benchmark functions available in the HPOLib package [1]: Support Vector Machine (SVM), Online Latent Dirichlet Allocation (LDA) and Logistic Regression, with, respectively, 3, 3, and 4 hyperparameters. In these datasets, all values were pre-computed for different configurations of hyperparameters. The number of observations are 1 400, 289, and 9 680 for each dataset respectively.

Next, we repeat the experiments of [7] for training a 1-hidden layer neural network and performing active learning for robot pushing.[1] Here we describe the former and in the next block the latter. Neural networks were trained to perform regression on the Boston housing dataset and classification with the breast cancer dataset [4]. The four tuned hyperparameters were the number of neurons, the damping factor $\mu$, the $\mu$-decrease factor, and the $\mu$-increase factor. The neural network's initial weights and all other parameters were set to be the same. The $L^2$-loss on the validation set of the Boston housing dataset is the objective function to be minimized, and the classification error on the validation set of the breast cancer dataset is the black-box function.

**Active learning for robot pushing** Bayesian optimization was used to do active learning for the pre-image learning problem for pushing [3]. The objective function to be optimized takes as input the pushing action of the robot, and outputs the distance of the pushed object to the goal location. The goal is to minimize the function in order to find a good pre-image for pushing the object to a given location. Two functions were tested: a 3-dimensional input with robot location $(r_x, r_y)$ and pushing duration $t_r$, and a 4-dimensional input which has the robot location and the angle $(r_x, r_y, r_\theta)$ and the pushing duration $t_r$. To facilitate reproducibility, we selected one goal location and precomputed 20 000 values for each test function. The inputs were random locations created using a low-discrepancy Sobol sequence and an appropriate mapping to the input domain.

**Determining Cosmological Parameters** Finally, we test our approach on the task of determining cosmological constants of standard physical models of the universe (e.g., Hubble's constant, dark energy density, matter density, etc.). The exact values of those constants are unknown but scientists can run simulations to estimate the likelihood of a particular setting of constants given our current experimentally observed data of the universe. Our goal is to find the set of constants (a total of 9 parameters) with maximum likelihood. We use the same data and software used by the authors of [2] and described in [6]. We also precomputed the objective function values but for an even larger set of locations, 50 000 in total.

Table 1: Test functions used in our experiments. The analytic form of these functions as well as the global minimum can be found online [5].

| test function | domain |
| --- | --- |
| Ackley | $[-5, 5]^d, d \in \{2, 5\}$ |
| Beale | $[-4.5, 4.5]^5$ |
| Branin | $[-5, 10] \times [0, 15]$ |
| Eggholder | $[-512, 512]^2$ |
| Six-Hump Camel | $[-3, 3] \times [-2, 2]$ |
| Drop-Wave | $[-5.12, 5.12]^2$ |
| Griewank | $[-600, 600]^d, d \in \{2, 5\}$ |
| Rastrigin | $[-5.12, 5.12]^2, d \in \{2, 4\}$ |
| Rosenbrock | $[-5, 10]^2$ |
| Shubert | $[-10, 10]^2$ |
| Hartmann | $[0, 1]^d, d \in \{3\}$ |
| Levy | $[-10, 10]^3$ |

(a) Robot pushing 4D      (b) Neural Network Cancer

Figure 1: Averaged (log) minimum observed function value and standard error of all methods for the remaining functions.

## Footnotes

[1]Code available online:https://github.com/zi-w/Max-value-Entropy-Search/