[Reviews · NeurIPS 2018]

Reviewer 1



The manuscript proposes a principled Bayesian technique for simultaneously optimizing a blackbox function and learning which is the most likely function class that explains the observations from the blackbox. The method supports a large class of functions by exploring the space of kernels via composition of two base kernels. It seems like the technique trivially extends to more base kernels and composition operations---or perhaps eventually prior mean functions. The experimental results show a significant benefit to intelligently searching the space of models (in this case kernels) when compared to fixed mixtures or models and random walks in model space. The paper is very clearly written and goes into an appropriate level of detail for a Bayesian optimization audience. If I had but one nitpick in this regard, it would be that the ABOMS algorithm would benefit from a pseudocode description; indeed, I am still unsure of the difference between ABO and ABOMS. The algorithm is principled and technically sound yet relatively simple to implement, which should make it accessible and impactful, especially with promised released code. The results show the performance and robustness of the algorithm when compared to random search, a fixed model, a fixed mixture of models, and a random walk through models; though the latter seems to work pretty well as well.

Reviewer 2



Since the efficiency of Bayesian optimization mostly hinges on properly modelling the objective function, picking an appropriate model is essential. Usually, the model of choice is a Gaussian process with a simple Matern or SQE kernel, which is prone to model miss-specification. The proposed method extends the main loop in Bayesian optimization by an additional step to automatically select promising models based on the observed data. To solve this inner optimization problem, they use Bayesian optimization in model space to find a composition of kernels that account for the uncertainty of the objective function. Overall I do think that the method is sensible and addresses an important problem of Bayesian optimization, i.e model miss specification. Also, the paper is well written and clearly structured and I really enjoyed reading it. However, even though the authors show that the proposed method outperforms a set of baselines, I am bit concerned that all the necessary approximations might make the method brittle against the choice of its own hyperparameters in practice. Questions for rebuttal: 1) Could you give some more insights how sensitive the method is against its hyperparameters? 2) How high is the additional optimization overhead of the method. Could you plot the performance also over wall-clock time? 3) Does the method also allow to combined models other than Gaussian processes, i.e random forest, Bayesian neural networks etc.? 4) Do you have any insights how the model changes during the optimization? I assume that the components of the ensemble become at some point relatively similar. 5) Does it make sense to combine the model selection with from the acquisition function to gather data that helps to obtain a better model? I think of an active learning phase in the beginning where Bayesian optimization is mostly exploring anyway. --- Post Rebuttal --- I thank the authors for answering my questions and I appreciate that they will clarify them in the supplementary material.

Reviewer 3



PAPER SUMMARY: This paper investigates a new Bayesian Optimization (BO) approach that automatically learns the correlation structure (i.e., kernel function) of the GP model of the black-box function while collecting observation to optimize it. The key idea is to run a second BO on the model space (i.e., a collection of complex kernels generated by summing and multiplying base kernels) at each iteration to update the list of model candidates before choosing the next observation. This 2nd BO routine casts the model evidence as a random function distributed by another GP. The implementation is adopted from a previous work (BOMS) on model selection via BO. SIGNIFICANCE & NOVELTY: This is an interesting problem & the approach taken is fairly new to me. I do, however, have the following concerns regarding the significance & technical soundness of the proposed solution: The authors propose to treat the data collected at earlier stages as noisier observations to avoid recomputing the model evidence (as new data arrives) & claim that it would make the search better informed. This is, however, a strange intuition: adding (implicitly) more noise to earlier observations only increases the prediction variance & while it might help preventing the model from being overly confident in its prediction, it does not make the model more informed because the quality of information remains the same if the model evidence is not recomputed. Section 4 (line 150-151) seems to suggest that the noise variance of a sample candidate depends on the previously collected samples. If this is indeed the case, the integration in line 179 is no longer analytically tractable. The description of Algorithm 1 is somewhat confusing: the update model sub-routine is said to recompute the corresponding model evidence before running the 2nd BO sub-routine on the model space but I thought the entire point here is to exploit heterogeneous noise scheme and avoid recomputing them? What exactly does the update model sub-routine do? How do we set the model prior p(M)? At the beginning of a new iteration, the pool of model candidate is expanded & the prior will need to be changed. This does not seem to be discussed in the main text. Also, another issue is the large no. of approximation layers involved in computing the acquisition function for the main BO routine (see Section 4.1). This might affect the solution quality and I am raising this concern because I think the experiments are not very conclusive in this regard (please see my comments below). EXPERIMENT: If I understand correctly, the whole point of adding an extra model-searching BO sub-routine is to boost the performance of the main BO process. To demonstrate this conclusively, besides the experiments that the authors have presented, the proposed approach should also be compared with other state-of-the-art BO algorithms (e.g., predictive entropy search (PES) [1], upper-confidence-bound (UCB) ant etc., [2]) implementing the best model among a set of pre-selected candidates. However, in this regard, the proposed approach is only compared with a baseline including a standard EI implementing the simple non-isotropic SE kernel. Is this SE model among the proposed approach’s pre-selected pool of candidates & is it the best among those? Also, the experiment description seems to suggest that this baseline is subject to the same amount of approximations in Section 4 (see lines 185-186). This is somewhat strange because EI is analytically tractable and such approximations are not necessary and only serve to weaken the baseline. For a more conclusive demonstration, I would suggest the authors to compare their method with other state-of-the-art BO algorithms (without model selection) implementing (a) the worst and (b) the best model among those in the pre-selected pool. If the model selection machinery is efficient, we should at least see ABO outperforms (a) and approaching (or surpassing) (b) & this would resolve the last issue that I mentioned in the previous section (regarding the stable quality of the resulting solution after being subjected to multiple approximations). CLARITY: The paper is clearly written. REVIEW SUMMARY: I find the problem investigated in this paper novel & interesting but I have a few concerns and doubts regarding the proposed solution as well as its empirical evaluations, which are not very conclusive. --- [1] https://arxiv.org/abs/1406.2541 [2] https://arxiv.org/pdf/1206.2944 --- Thank you for the rebuttal. My first comment in the significance & novelty section is about the inner BO routine (not the outer BO routine as interpreted in your response). I do understand that by adding noises to the evidence observation to discourage poor model & at the same time, acknowledge that some of the old model evaluations might be out of date. This however does not really explain how this would make the inner BO routine more informed in model search (e.g., how would a previously discarded model get reconsidered in light of new data? If the model search only uses its model evaluation + artificial noise, how could the predicted score of the discarded model be improved to make the search more informed?). Secondly, part of my concern is whether the SE model is competitive against other model candidates in the pool. If it is less competitive than the others then the benefit from considering more model is not shown conclusively. I still think that the authors should compare with other models in the pool to demonstrate the efficiency of the developed model selection machinery.